# Triacylglycerol synthesis enhances macrophage inflammatory function

Angela Castoldi[1], Lauar B. Monteiro[1,2], Nikki van Teijlingen Bakker[1,4], David E. Sanin[1], Nisha Rana[1], Mauro Corrado[1], Alanna M. Cameron[1], Fabian Hässler[1], Mai Matsushita[1], George Caputa[1], Ramon I. Klein Geltink[1], Jörg Büscher[3], Joy Edwards-Hicks[1], Erika L. Pearce[1] & Edward J. Pearce[1,4✉]

Foamy macrophages, which have prominent lipid droplets (LDs), are found in a variety of disease states. Toll-like receptor agonists drive triacylglycerol (TG)-rich LD development in macrophages. Here we explore the basis and significance of this process. Our findings indicate that LD development is the result of metabolic commitment to TG synthesis on a background of decreased fatty acid oxidation. TG synthesis is essential for optimal inflammatory macrophage activation as its inhibition, which prevents LD development, has marked effects on the production of inflammatory mediators, including IL-1β, IL-6 and PGE2, and on phagocytic capacity. The failure of inflammatory macrophages to make PGE2 when TG-synthesis is inhibited is critical for this phenotype, as addition of exogenous PGE2 is able to reverse the anti-inflammatory effects of TG synthesis inhibition. These findings place LDs in a position of central importance in inflammatory macrophage activation.

[1] Department of Immunometabolism, Max Planck Institute of Epigenetics and Immunobiology, Freiburg im Breisgau, Germany. [2] Laboratory of Immunometabolism, Department of Genetics, Evolution, Microbiology and Immunology, University of Campinas, Campinas, São Paulo, Brazil. [3] Metabolomics Facility, Max Planck Institute of Epigenetics and Immunobiology, Freiburg im Breisgau, Germany. [4] Faculty of Biology, University of Freiburg, Freiburg im Breisgau, Germany. ✉email: pearceed@ie-freiburg.mpg.de

Macrophages are integral to most tissues. Foam cells, macrophages with lipid droplets (LDs) which are stores of triacylglycerols (TGs) and cholesterol esters (CEs), are found in various disease states[1]. LDs can act as energy stores since TG lipolysis releases fatty acids (FAs) for mitochondrial oxidation (FAO), a process that relies on long-chain FA conversion into acylcarnitines by the enzyme Cpt1a[2]. However, in macrophages, proinflammatory signals result in diminished FAO and increased TG synthesis with LD development[3,4].

Inflammation has critical protective functions, but when unregulated can also cause disease, in which IL-1β and other proinflammatory cytokines are implicated[5]. Recent work has established a strong link between inflammatory activation signals and induced changes in macrophage metabolism that are essential for the cells to perform their subsequent functions[6]. Major components of this metabolic reprogramming include enhanced Warburg metabolism, associated with diminished oxidative phosphorylation (OXPHOS) and altered TCA cycle dynamics in which glucose carbon is redistributed via citrate/aconitate to the synthesis of FAs and itaconate[7]. Increased FA synthesis in inflammatory macrophages is associated with additional broad changes in lipid metabolism[8], including the accumulation of TGs and CEs[3,9], which are stored within LDs. TG synthesis depends on the acyl-CoA:diacylglycerol acyltransferases 1 and/or 2 (DGAT1,2)[10], which catalyze the covalent addition of a fatty acyl chain to diacylglycerol (DG)[10]. LDs are the core energy storage organelles of adipocytes, but develop in other cell types as well, where they can again act as energy stores for fueling cell intrinsic ATP production via mitochondrial FAO[2,11]. However, LDs are recognized to mediate additional functions including the sequestration of toxic lipids and the prevention of excessive endoplasmic reticulum (ER) stress, and as lipid donors for autophagosome formation[2,12,13]. Of relevance for this study, there is also a body of literature which demonstrates that LDs act as a platform for eicosanoid production from the lipid substrate, arachidonic acid (AA)[14,15].

In vivo, LD-containing macrophages are most well recognized in atherosclerotic lesions and in tuberculosis[1,16]. More recent work has highlighted the presence of foam cells in active multiple sclerosis, certain cancers, white adipose tissue during obesity, and in bronchoalveolar lavage from individuals suffering from vaping-related lung disease[1]. The ratio of TGs to CEs in LDs in different settings is likely to be important, and there is ongoing discussion regarding whether macrophages that contain CE-rich LDs are universally proinflammatory[17,18].

Here we explore the significance of LDs in inflammatory macrophages. Macrophages stimulated with lipopolysaccharide (LPS) plus interferon-γ (IFNγ) accumulate TGs in LDs, and long-chain acylcarnitines. In these cells, inhibition of TG synthesis results in diminished LD development, and increased long-chain acylcarnitine levels, suggesting that FA fate is balanced between TG and acylcarnitine synthesis. Nevertheless, TG-synthesis is required for inflammatory macrophage function, as its inhibition negatively affects production of proinflammatory IL-1β, IL-6, and PGE2, and phagocytic capacity, and protects against LPS-induced shock in vivo. Failure to make PGE2 is critical for this phenotype, as exogenous PGE2 reverses the anti-inflammatory effects of TG-synthesis inhibition.

Our findings support the view that LDs are an integral consequence of a range of metabolic events that are initiated by LPS/IFNγ signaling, and that they have an important function in PGE2 synthesis, which is essential for maximal activation. This interpretation of LD function in inflammatory macrophages brings these organelles into focus as a potential therapeutic target for inflammatory disorders linked to excessive inflammatory mediator production.

## Results

### Fatty acid metabolism reprogrammed in inflammatory macrophages.
Inflammatory macrophages increase the commitment of resources to fatty acid synthesis[7]. Consistent with this, these cells accumulate LDs[3,9], but the functional significance of LDs in this setting is unclear, especially since inflammatory macrophages do not use FAO[3,4]. We began to address this by investigating the process and functional significance of TG synthesis in response to the proinflammatory stimulus provided by LPS plus IFNγ (Fig. 1a). We found that the expression of both *Dgat1* and *Dgat2*, the enzymes responsible for synthesizing TGs from DGs, increased in inflammatory macrophages as part of a broader transcriptional program including genes involved in TG synthesis and utilization (Fig. 1a), but that *Dgat1* was most strongly expressed (Supplementary Fig. 1a). *Gpat3*, which encodes the enzyme which catalyzes the conversion of glycerol-3-phosphate to lysophosphatidic acid in the synthesis of TG, was the TG synthesis pathway gene most upregulated as a result of inflammatory activation (Fig. 1a, Supplementary Fig. 1b). As expected, TG and LD accumulation were also marks of inflammatory activation (Fig. 1b–d; Supplementary Fig. 1c). Consistent with their use for TG synthesis (Fig. 1a), overall levels of free FA and DGs were diminished in inflammatory compared to resting macrophages (Fig. 1e). Lipidomics revealed broad, dynamic changes in lipids as a result of inflammatory activation, with the accumulation of phosphatidylcholines (PCs), phosphatidylinositols (PIs), phosphatidylehtanolamines (PEs), lysophosphotidylcholines (LPCs), sphingomyelins (SMs), CEs and hexosylceramides (HEXCERs) beginning as early as 2 h post activation (Supplementary Fig. 1d). This was not the case in IL-4-stimulated alternatively activated macrophages (Supplementary Fig. 1d), which actively fuel FAO by lipolysis and which, unlike inflammatory macrophages, do not contain visible LDs[4]. Despite the fact that FAO and OXPHOS are inhibited during inflammatory activation[3,19], expression of *Cpt1a*, which encodes carnitine palmitoyltransferase 1 (Cpt1a), was found to be increased in inflammatory macrophages (Supplementary Fig. 1e). Cpt1a is located in the outer mitochondrial membrane where it converts long chain FA into acylcarnitines that can then be transported into the mitochondrial intermembrane space prior to subsequent transfer into the matrix for entry into FAO. We found that increased Cpt1a was associated with a significant drop in carnitine and accumulation of long chain acylcarnitines (Fig. 1f, Supplementary Fig. 1f). This was despite parallel increases in expression of *Cpt2*, which encodes the enzyme that catalyzes the removal of carnitine from acylcarnitines (Supplementary Fig. 1e). These changes did not correlate with utilization of FA in FAO, since while incorporation of $^{13}$C from labeled palmitate into C16 acylcarnitine was increased in inflammatory vs. resting macrophages, its incorporation into the TCA cycle intermediate citrate was diminished (Fig. 1g). This is consistent with previous findings that FAO declines in inflammatory macrophages. Furthermore, in inflammatory compared to resting macrophages $^{13}$C from $^{13}$C-labelled glycerol was preferentially incorporated into the TG precursor glycerol phosphate, rather than being used in glycolysis, where it is measurable as labeled 3-phosphoglycerate (Supplementary Fig. 1g). Taken together, our data support the view that inflammatory macrophages divert available resources toward the conversion of FA to TG which are stored in LDs, or to the synthesis of long-chain acylcarnitines (Fig. 1h).

### DGAT1 promotes TG accumulation and inflammatory capacity.
To ascertain the role of TGs in inflammatory macrophages, we used T863, a selective DGAT1 inhibitor (DGAT1i)[20] to suppress TG accumulation (Fig. 2a, b; Supplementary Fig. 2a, b). This

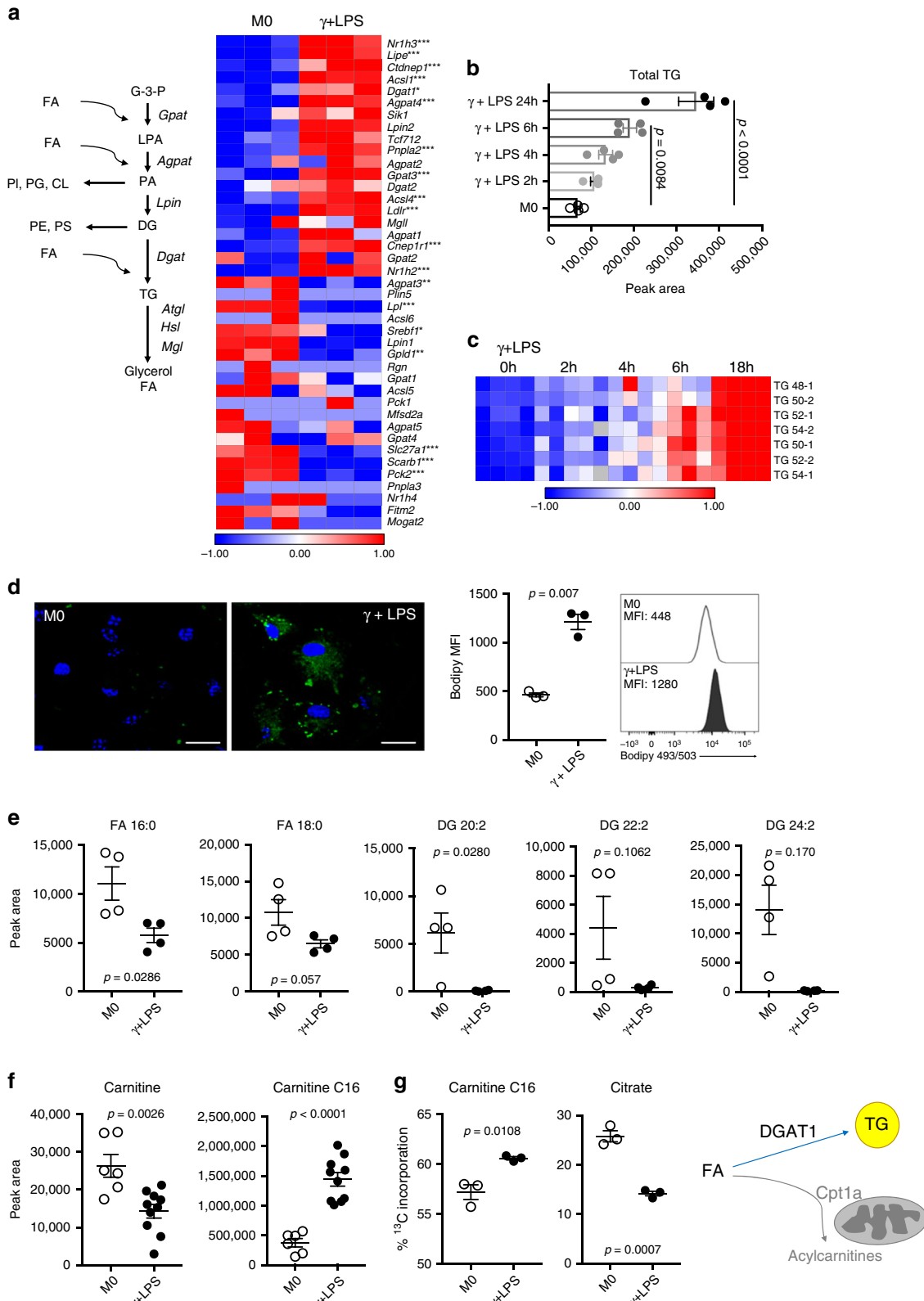

was accompanied by diminished LD accumulation (Fig. 2c), correlated with reduced staining with bodipy 493/503 (Fig. 2d; Supplementary Fig. 2c), and diminished C16 bodipy uptake (Fig. 2e; Supplementary Fig. 2d). DGATi also resulted in reduced levels of CEs and cardiolipins, and certain species of HEXCERs, PCs, PSs, and SMs, although it had no effect on PIs and caused increases in LPCs (Supplementary Fig. 2a). DGAT1i also caused the accumulation of substrate DGs in inflammatory macrophages (Fig. 2f; Supplementary Fig. 2e), but free FA levels were unaffected (Supplementary Fig. 2f), presumably because these were still being used to synthesize DGs. Suppression of DGAT1 using shRNA (Supplementary Fig. 2g) resulted in effects that were similar to those mediated by DGAT1i, with diminished TG accumulation (Fig. 2g) and staining with bodipy 493/503 (Fig. 2h;

**Fig. 1 Inflammatory activation results in the remodeling of fatty acid metabolism. a** TG biosynthetic pathway and expression of pathway genes, as measured using RNAseq, in resting (M0) and inflammatory macrophages (stimulated with IFNγ + LPS for 18 h) ($n = 3$ biologically independent samples). **b** Time-dependent accumulation of total TG, and **c** different TG species, in macrophages after IFNγ + LPS stimulation ($n = 4$ biologically independent samples). **d** Bodipy 493/503 staining in resting (M0) macrophages and macrophages stimulated with IFNγ + LPS, for 18 h, visualized by confocal microscopy and measured by flow cytometry (MFI, median fluorescence intensity). Scale bars represent 10 μm ($n = 3$ biologically independent samples). **e** Fatty acids (FA), and diacylglycerols (DG) ($n = 4$ biologically independent samples), **f** carnitine and C16 acylcarnitine (M0 $n = 4$; γ + LPS $n = 10$ biologically independent samples), and **g** total fraction contribution (FC) of palmitate carbons into C16 acylcarnitine and citrate. Resting (M0) macrophages and macrophages stimulated with IFNγ + LPS were cultured in the presence of 100 μM of $^{13}$C-palmitate for 18 h ($n = 3$ biologically independent samples). **h** Inflammatory macrophages convert FA to TG, which are stored in LDs (blue arrow), or to acylcarnitines through Cpt1a in the outer mitochondrial membrane (gray arrow). This manuscript is focusing on the significance of TGs for inflammatory macrophage activation. Data are represented as mean values ± s.e.m. One-way ANOVA with Bonferroni's multiple comparison test (**c**); Unpaired two-tailed Student's t test (**d–g**). (*$p < 0.05$, **$p < 0.01$, ***$p < 0.001$). **a** representative of one experiment, **b–f** representative of two experiments and **g** representative of tree experiments. Source data are provided as a Source Data file.

Supplementary Fig. 2h). We next queried the effects of the loss of DGAT1 function on inflammatory activation. We found that the failure to make TGs and thereby accumulate LDs in DGAT1i-treated or *Dgat1*-shRNA-transduced LPS/IFNγ-stimulated cells was accompanied by a significant reduction in inflammatory capacity, as measured by IL-1β (Fig. 2i, j; Supplementary Fig. 2i, j) and IL-6 production (Fig. 2k, l; Supplementary Fig. 2k). *Il1b* and *Il6* mRNAs were also diminished by DGAT1i (Supplementary Fig. 2l). IL-6 is known to be regulated by IL-1β[5] and thus reduced levels of IL-6 in these assays may be a downstream effect of inhibition of IL-1β production. These data are consistent with a previous report that DGATi diminished inflammatory cytokine production in *Mycobacterium tuberculosis* infected THP1 cells (a human macrophage cell line)[21]. In addition, we observed marked reductions in mRNAs encoding the Macrophage Inflammatory Proteins 1α (*Ccl3*) and β (*Ccl4*), the inflammasome Nalp1 (*Nlrp1b*), and reductions in expression of additional genes involved in the inflammatory response including *Ido1* and *Il1a* in inflammatory macrophages as a result of DGAT1 inhibition (Supplementary Fig. 2l), attesting to the proinflammatory effects of TG synthesis. The reduction in *Il1a* expression is of interest since IL-1α was strongly implicated in the pathophysiology of atherosclerosis associated with foamy macrophage development in oleate-fed mice[22]. Earlier work showed that interfering with LD dynamics through the inhibition of TG hydrolysis negatively impacted the ability of macrophages to engage in phagocytosis[23]. Here, we observed reduced phagocytic ability following the loss of DGAT1 function (Fig. 2m; Supplementary Fig. 2m).

We found that inhibiting DGAT1 also had significant effects on inflammation in vivo. T863 treatment of mice with LPS-induced systemic inflammation significantly diminished disease severity, which was measured as the hypothermic response to i.p. LPS injection (Fig. 2n). Mice that had received the drug were active and appeared unaffected by LPS injection at the time of sacrifice (10 h post injection), whereas those that had received carrier alone exhibited sickness signs, such as hunching, immobility, and piloerection. While in these experiments we cannot rule out systemic effects of T863 injection on cells other than macrophages, post mortem analysis of peritoneal macrophages revealed lower bodipy 493/503 staining in resident (F4/80$^+$ TIM4$^+$) and recruited (F4/80$^+$ TIM4$^{neg}$) macrophages (Fig. 2o, p; Supplementary Fig. 2n), indicative of lower LD formation. Thus i.p. T863 does impact the local macrophage response to LPS. Moreover, T863-treatment resulted in significantly decreased serum levels of the typical macrophage inflammatory products IL-1β (Fig. 2q) and IL-6 (Fig. 2r). These results are supportive of our in vitro findings and taken together the data indicate that the synthesis of TGs and their storage in LDs supports inflammatory macrophage activation.

**DGAT1 affects inflammatory macrophage metabolism.** Macrophage activation driven by IFNγ plus LPS is linked to increased aerobic glycolysis and the diversion of the TCA cycle intermediates citrate/aconitate to the production of itaconate[6,7]. We reasoned that reduced inflammatory capacity associated with the loss of DGAT1 function might be linked to alterations in one or more of these parameters. However, we found that inhibition of TG synthesis had no significant effect on glucose carbon incorporation into lactate (a downstream product of aerobic glycolysis) (Fig. 3a) or citrate (Fig. 3b) or on extracellular acidification rate (ECAR, a measure of released lactate) (Fig. 3c, Supplementary Fig. 3a) or itaconate production (Fig. 3d, Supplementary Fig. 3b) in DGAT1i-treated or *Dgat1*-shRNA-transduced LPS/IFNγ-stimulated cells. We also assessed mitochondrial function. We found that inner membrane potential (a mark of the proton gradient across the inner membrane) (Fig. 3e; Supplementary Fig. 3c) and mitochondrial mass (Fig. 3f; Supplementary Fig. 3d) were equivalent or greater, respectively, in cells that lacked DGAT1 function, and that mitochondrial reactive oxygen species (ROS) were increased (Fig. 3g, Supplementary Fig. 3e, f). Consistent with diminished respiration in inflammatory macrophages[19], ultrastructural analyses revealed that mitochondrial cristae in these cells were looser than in resting cells[24,25], but this parameter was not obviously affected by DGAT1 inhibition (Fig. 3h). As expected, LDs were evident in inflammatory macrophages to a greater extent than in resting macrophages (Supplementary Fig. 3g), and these organelles were less frequent in cells treated with DGAT1i (Fig. 3h, i; Supplementary Fig. 3g).

In other cell types, DGAT1 inhibition has been associated with increased ROS and increased long chain acylcarnitine accumulation[26]. Similarly, we found increased long chain acylcarnitine pools (Fig. 3j, Supplementary Fig. 3h), associated with diminished pools of carnitine (Fig. 3j) and correlated with increased mitochondrial ROS (Fig. 3g, Supplementary Fig. 3e, f) as a result of loss of DGAT1 function in inflammatory macrophages. However, these changes were not indicative of increased mitochondrial stress, as expression of mitochondrial-unfolded-protein response genes *Dnaja3*, *Hspa9*, and *Hspd1* was not increased upon DGAT1 inhibition (Fig. 3k, Supplementary Fig. 3i) in LPS plus IFNγ-stimulated macrophages.

TGs are synthesized in the ER, which we have previously reported is significantly expanded in dendritic cells activated by LPS[27]. We found a similar effect in macrophages following stimulation with LPS plus IFNγ (Supplementary Fig. 3k). Whereas in DCs ER expansion was shown to require fatty acid synthesis, and to support increased cytokine production, here we found that despite causing diminished cytokine production, inhibition of TG synthesis resulted in an additional increase in ER (Fig. 3l, Supplementary Fig. 3j, k). We speculate that this may reflect the accumulation of intermediates of the TG synthesis

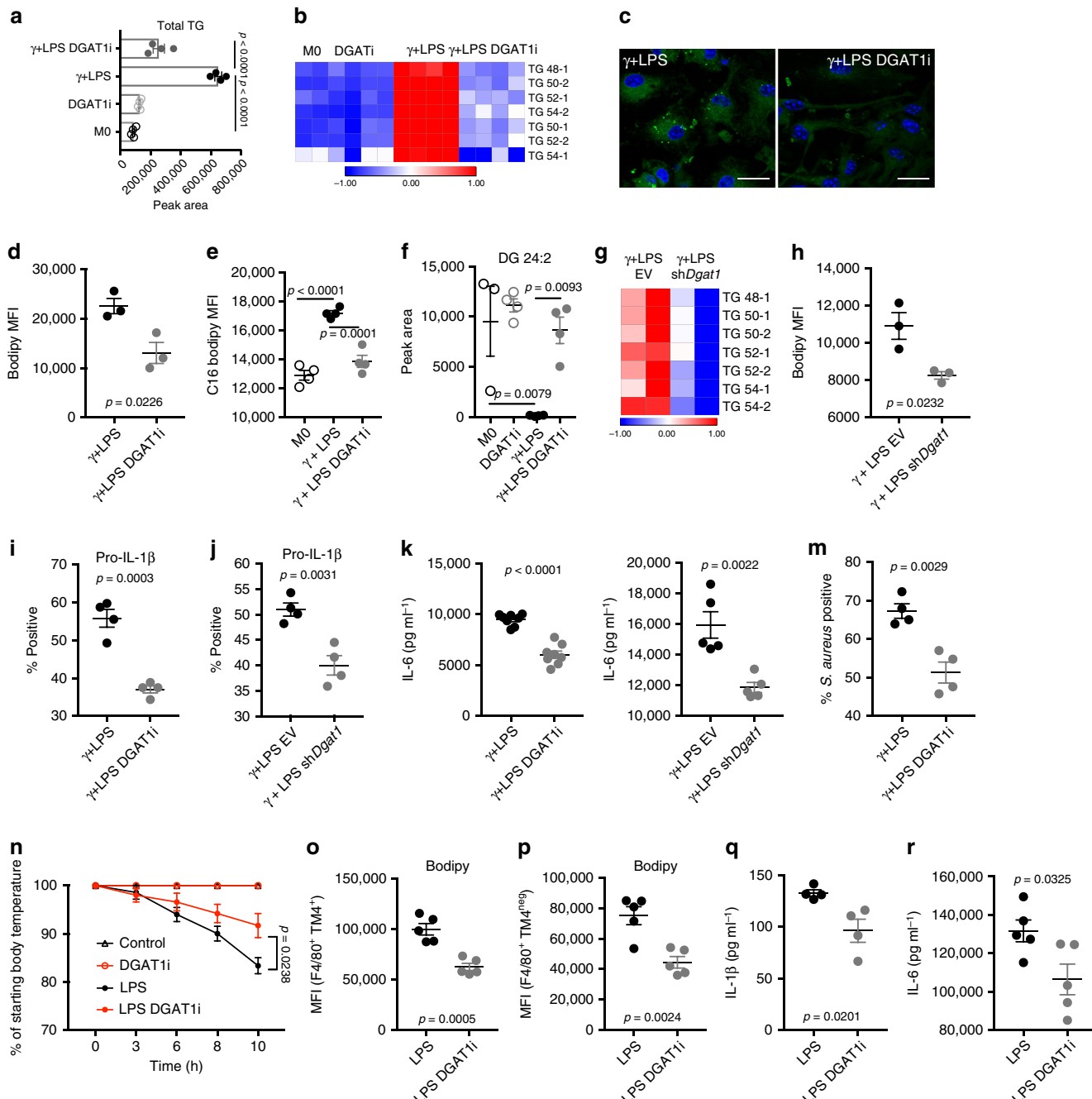

**Fig. 2 The loss of DGAT1 function decreases TG accumulation, LDs, and inflammatory function. a** Effect of DGAT1 inhibitor T863 (DGAT1i) on total TG, and **b** distinct TG species ($n = 4$ biologically independent samples) in resting (M0) and macrophages stimulated with IFNγ + LPS. **c** Bodipy 493/503 fluorescence visualized by fluorescence microscopy (scale bars represent 10 μm) or **d** measured by flow cytometry (MFI, median fluorescence intensity) ($n = 3$ biologically independent samples). **e** C16 Bodipy uptake ($n = 4$ biologically independent samples). **f** Effect of DGAT1i on DG levels in M0 or macrophages stimulated with IFNγ + LPS ($n = 4$ biologically independent samples). **g** Effect of Dgat1-shRNA (shDgat1) or empty vector (EV) on TG species. **h** Bodipy 493/503 fluorescence in macrophages transduced with shDgat1 or EV and treated with IFNγ + LPS ($n = 3$ biologically independent samples). **i** Pro IL-1β positive macrophages stimulated with IFNγ + LPS, plus/minus DGATi or **j** shDgat1 ($n = 4$ biologically independent samples). **k, l** Effect of the loss of DGAT1 function on IL-6 production (k $n = 8$ and l $n = 5$ biologically independent samples). **m** Effect of DGAT1i on phagocytosis of *S. aureus* (PE-labelled) ($n = 4$ biologically independent samples). **n** C57BL/6 mice were injected with DGAT1i, or carrier, 30 min before receiving 8 mg kg$^{-1}$ of i.p. LPS or PBS. Drop in body temperature was calculated in relation to the initial body temperature. **o** Bodipy 493/503 fluorescence in large (F4/80$^{+}$TIM4$^{+}$), and **p** small (F4/80$^{+}$TIM4$^{neg}$), peritoneal macrophages from the mice mice shown in (**n**), recovered at 10 h post injection of LPS or PBS ($n = 5$ biologically independent samples). **q** Serum IL-1β at 2 h post injection ($n = 4$ biologically independent samples). **r** Serum IL-6 at 10 h post injection ($n = 5$ biologically independent samples). For **a–g** and **k–m**, assays were at 18 h post activation; **i, j** at 6 h. Data are represented as mean values ± s.e.m. One-way ANOVA with Bonferroni's multiple comparison test (**a, e, f, n**); Unpaired two-tailed Student's *t* test (**d, h–m, p–r**). (**a, b, e–h, n–r**) representative of two or (**c, d, i–m**) three experiments. Source data are provided as a Source Data file.

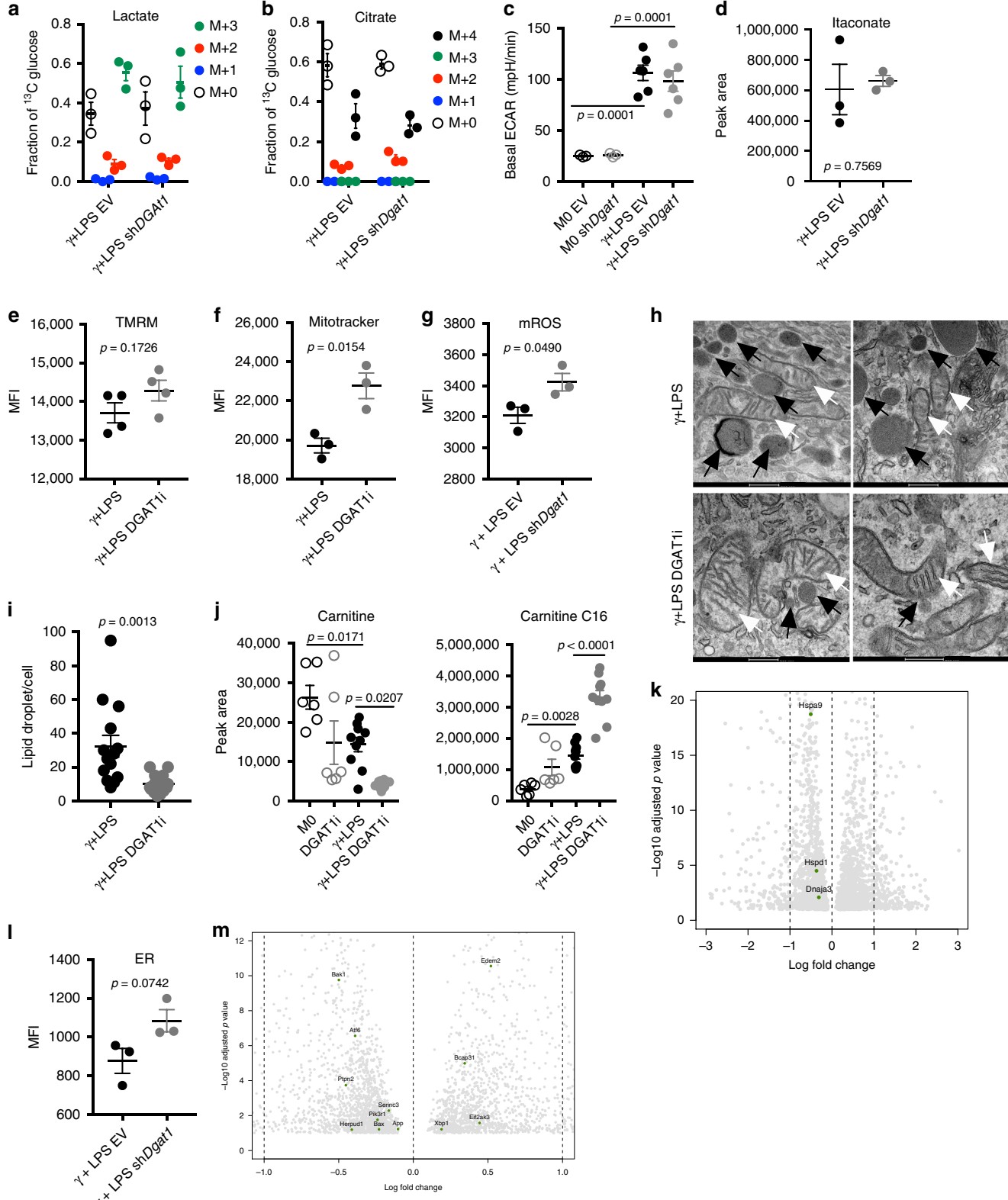

pathway in the ER. However, a lack of transcriptional response within the unfolded-protein response genes *Xbp1, Ern1, Ddit3, Atf6, Hspa5, Eif2ak3, Atf4*, and *Cebpb* indicated that DGAT1 loss of function did not cause ER stress in inflammatory macrophages (Fig. 3m, Supplementary Fig. 3l).

**TG synthesis regulates PGE2**. As a means to understand underlying changes in inflammatory function, we explored the transcriptional signature associated with the loss of DGAT1 function. Inhibition of DGAT1 resulted in significant changes in expression of 285 genes (Supplementary Fig. 4a). Amongst the six most significantly downregulated pathways within this set, three were linked to eicosanoids, with two specifically associated with the regulation of prostaglandin secretion (Fig. 4a, b). PGE2 is produced by inflammatory macrophages, in which expression of prostaglandin synthase *Ptges2* was elevated compared to in

**Fig. 3 The lack of effect of the loss of function of DGAT1 on core metabolism in inflammatory macrophages. a** Fraction contribution of glucose carbons into lactate and **b** citrate, in macrophages transduced with EV or sh*Dgat1*, stimulated with IFNγ + LPS for 18 h and cultured in $^{13}$C-glucose for the entire time ($n = 3$ biologically independent samples). **c** Basal ECAR of transduced macrophages cultured with IFNγ + LPS or without (M0) for 18 h (M0 $n = 3$, γ + LPS $n = 6$ biologically independent samples). **d** Total pool of itaconate in transduced macrophages cultured with IFNγ + LPS or without (M0) for 18 h ($n = 3$ biologically independent samples). **e** Effects of DGAT1 loss of function on mitochondrial membrane potential ($n = 4$ biologically independent samples), **f** mitochondrial mass and **g** mitochondrial reactive oxygen species after stimulation with IFNγ + LPS for 18 h ($n = 3$ biologically independent samples), (MFI, median fluorescence intensity). **h** Electron microscopy of macrophages at 18 h post stimulation with IFNγ + LPS without or with DGAT1i. Black arrows, LDs; white arrows, mitochondria (scale bars represent 500 nm). **i** LD/cell count fom electron microscopy ($n = 14$ cells from 3 biologically independent samples). **j** Carnitine and C16-acylcarnitine in resting (M0) or IFNγ + LPS-stimulated macrophages, cultured with or without DGAT1i for 18 h (M0 $n = 6$, DGAT1i $n = 6$, γ + LPS $n = 10$, γ + LPS DGAT1i $n = 10$ biologically independent samples). **k** Mitochondrial unfolded protein response genes at 18 h post stimulation with IFNγ + LPS without or with DGAT1i as measured using RNAseq (Log2 Fold change). **l** Endoplasmic reticulum (ER) mass in transduced macrophages stimulated with IFNγ + LPS for 18 h ($n = 3$ biologically independent samples). **m** ER stress response genes at 18 h post stimulation with IFNγ + LPS without or with DGAT1i as measured using RNAseq (Log2 Fold change). Data are represented as mean values ± s.e.m. One-way ANOVA with Bonferroni's multiple comparison test (**a**–**c**, **j**); Unpaired two-tailed Student's t test (**d**, **e**, **g**, **i**, **l**). **h**–**i**, **k**, **m** representative of one experiment, **a**–**g**, **j**, **l** representative of three experiments. Source data are provided as a Source Data file.

resting macrophages (Supplementary Fig. 4b). Furthermore, COX2 (encoded by *Ptgs2*) is upregulated in inflammatory macrophages compared to unstimulated macrophages (Supplementary Fig. 4b). In addition, expression of additional genes that regulate PGE2 synthesis, such as *Il1b*, *Ptges*, *Mgst1*, and *Nod2* were significantly upregulated, whereas *Ptgs1*, together with *Itgam* and *Cbr1* were downregulated (Supplementary Fig. 4b).

Recent findings have implicated autocrine effects of PGE2 in the production of IL-1β [28]. Given that LDs are both TG stores and sites for PGE2 synthesis, and the loss of DGAT1 function resulted in loss of LDs and of IL-1β production, we asked whether this was associated with reduced PGE2 levels. We found that PGE2 production was significantly increased in response to inflammatory signals, and consistent with a role for TGs in PGE2 synthesis, suppressed by DGAT1 inhibition (Fig. 4c). Moreover, we found that inflammatory macrophages build FA 20:4 (AA)-containing TG stores, which serve to focus substrate for PGE2 synthesis within these organelles. The inhibition of TG synthesis prevented the accumulation of this substrate store (Fig. 4d and Supplementary Fig. 4c). The cells also contained FA 20:4 (AA) in PC, PE, PG and PS (Supplementary Fig. 4d), but for the most part these were not dynamically regulated by activation (Supplementary Fig. 4d). We next tested whether exogenous PGE2 was able to promote inflammatory cytokine production in the absence of DGAT1. Exogenous PGE2 increased pro IL-1β and IL-6 production by inflammatory macrophages in the presence of DGAT1i (Fig. 4e, f; Supplementary Fig. 4e, f) or *Dgat1*-shRNA (Supplementary Fig. 4g, h). There was also a proinflammatory effect of exogenous PGE2 in cells in which DGAT1 was functional (Fig. 4e, f; Supplementary Fig. 4g, h), but the PGE2-induced fold increase in cytokine production was significantly greater in cells in which TG synthesis was inhibited (Fig. 4g, Supplementary Fig. 4i). We next asked whether PGE2 could also rescue the loss of phagocytic ability associated with inhibited TG synthesis. We found that the addition of exogenous PGE2 prevented the decline in phagocytosis-competent cells associated with the loss of DGAT1 function (Fig. 4h). Moreover, PGE2 increased the intrinsic phagocytic capacity of both control and *Dgat1*-shRNA-transduced inflammatory macrophages (Supplementary Fig. 4j). Thus, our results indicate that inflammatory macrophages require increased TG synthesis and LD formation in order to allow the synthesis of PGE2 which provides a second signal for inflammatory activity.

## Discussion

Altered mitochondrial function, in which FAO and OXPHOS are inhibited and the TCA cycle is fragmented to support the use of

citrate and aconitate for fatty acid and itaconate production respectively, are core metabolic features of inflammatory macrophage activation [3,6,7,29–31]. These events, alongside increased FA uptake and increased DGAT expression, create an environment in which TG synthesis is promoted. The finding that LDs, which store TGs, play an important role in maximizing inflammatory macrophage function raises the possibility that LD development is a major objective of metabolic rewiring in these cells.

Our data indicate that an activation induced increase in DGAT1 expression plays a dominant role in increased TG synthesis, while DGAT2 plays only a supportive role. Although we cannot exclude the possibility that the DGAT1 inhibitor T863 has off-target effects that impact inflammatory macrophage biology, data generated from experiments in which DGAT1 expression was suppressed by RNAi support the conclusion that DGAT1 is playing a critical role in the production of TGs and LDs in inflammatory macrophages. In both loss of function models, decreased TG synthesis and LD formation was associated with impaired inflammatory capacity. Our data are compatible with previous reports that DGAT1 inhibition is sufficient to reduce TG levels in macrophages and other cell types [32,33].

A surprising finding from the current studies was that, correlated with increased *Cpt1a* expression, inflammatory macrophages have increased pools of long chain acylcarnitines. We reason that this reflects increased substrate levels due to increased FA uptake and synthesis, and subsequent acylcarnitine accumulation due to mitochondrial dysfunction with the cessation of FAO. The functional significance of acylcarnitine accumulation is unknown, but given that long-chain acylcarnitines increased further when TG synthesis was inhibited in inflammatory macrophages, it may serve as a default alternative pathway to LD development (Fig. 1h) for sequestering FA. Increased acylcarnitine accumulation following DGAT1 inhibition has been reported previously [32,34]. However, in MEFs, accumulated acylcarnitines due to DGAT1 inhibition were reported to themselves cause mitochondrial dysfunction [32]. Export of long chain acylcarnitines from mitochondria and from cells has been reported [35], raising the possibility that accumulated acylcarnitines may be released to serve specific extracellular functions, including roles in sterile inflammation [36] associated with macrophage activation. The dissociation of increased *Cpt1a* expression from FAO is important, since Cpt1a/Cpt2-dependent FAO generally works to counteract LD development in macrophages [37], and enforced expression of a permanently active Cpt1a mutant in a macrophage cell line (RAW) increased FAO and subsequently prevented LD development and inhibited cytokine production in response to an inflammatory signal [38].

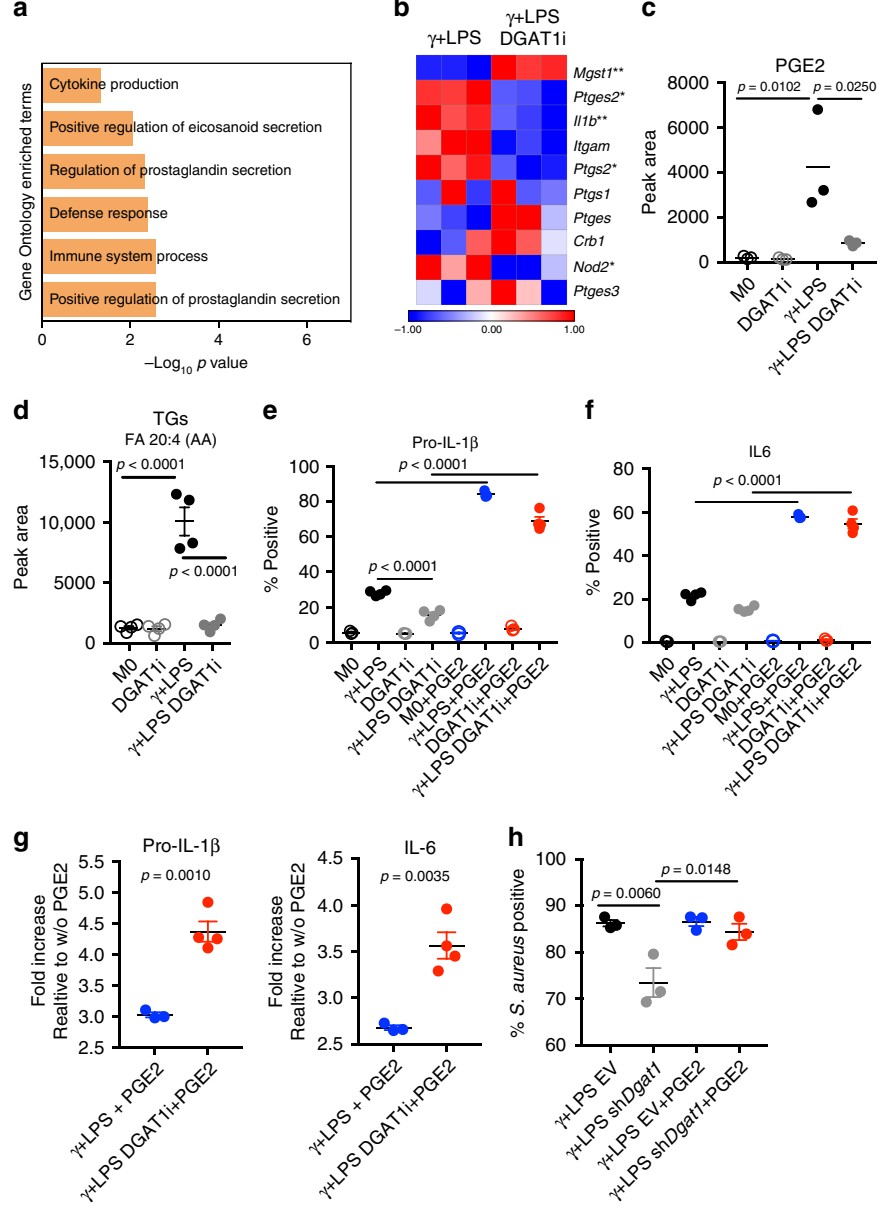

**Fig. 4 Reduced PGE2 production downstream of inhibited TG synthesis affects inflammatory macrophage function. a** Down regulated enriched pathways in macrophages stimulated with IFNγ + LPS in the presence or absence of DGAT1i for 18 h. Assessed using RNAseq data. **b** Prostaglandin-E synthase activity genes expression at 18 h post stimulation with IFNγ + LPS without or with DGAT1i as measured by RNAseq. **c** Total pool of prostaglandin E2 (PGE2) in resting (M0) or IFNγ + LPS-stimulated macrophages, treated with or without DGAT1i for 18 h ($n = 3$ biologically independent samples). **d** TGs containing FA 20:4 (arachidonic acid) at 18 h in M0, or macrophages stimulated with IFNγ + LPS, with or without DGATi ($n = 4$ biologically independent samples). pro IL-1β (**e**) or IL-6 (**f**) positive macrophages (percentage cytokine-positive F4/80+ cells, measured by flow cytometry), 6 h after stimulation with IFNγ + LPS with or without DGATi and/or exogenous PGE2, as shown. PGE2 was added 1 h after IFNγ + LPS or IFNγ + LPS + DGATi, and cells were collected 5 h later ($n = 3$ for controls without γ + LPS and $n = 4$ for γ + LPS biologically independent samples). **g** Fold increases in pro IL-1β and IL-6 production (measured as percentage cytokine-positive F4/80+ cells) induced by exogenous PGE2 in macrophages stimulated with IFNγ + LPS in the presence or absencer of DGAT1i (γ + LPS + PGE2 $n = 3$, γ + LPS DGAT1i + PGE2 $n = 4$ biologically independent samples). **h** Effect of PGE2 on the ability of EV or shDgat1-transduced macrophages to phagocytose S. aureus (PE-labelled) after simulation with IFNγ + LPS. Data show percentage of F4/80+ cells positive for S.aureus ($n = 3$ biologically independent samples). Data are represented as mean values ± s.e.m. One-way ANOVA with Bonferroni's multiple comparison test (**c-f**, **h**); Unpaired two-tailed Student's t test (**g**). (*$p < 0.05$, **$p < 0.01$). (**a**, **b**, **d**) representative of one experiment, (**c**, **e-h**) representative of two experiments. Source data are provided as a Source Data file.

Our findings indicate that TG synthesis is necessary for the accumulation of AA-containing TGs to act as a substrate reservoir for PGE2 synthesis. In this scenario, coordinated increases in LD development and expression of genes related to positive regulation of prostaglandin-E synthase activity enable the framework for enhanced PGE2 synthesis, which provides a strong positive feedback signal for the production of pro-IL-1β[28]. IL-1β is considered the gatekeeper of inflammation in both infectious and sterile settings, since it is able to broadly induce the production of other proinflammatory mediators such as IL-6, TNF, GM-CSF, G-CSF, IL-8, NO, as well as PGE2 and additional IL-1β[39]. While events that we have shown here to be linked to each

other through LDs, such as IL-1β and PGE2 production, have been linked to protective mechanisms in microbial infections[40–42], there are also clear examples in which exacerbated production of these mediators is linked to the development of severe diseases such as cancer, neurodegenerative diseases, atherosclerotic disease, and rheumatologic disorders[43–47]. Our findings raise the possibility of targeting TG synthesis for therapeutic purposes in these settings.

## Methods

**Mice and in vivo experiments**. C57BL/6 mice (RRID: IMSR_JAX:000664) were from The Jackson Laboratory, and were maintained in specific pathogen-free conditions under protocols approved by the animal care committee of the Regierungspräsidium Freiburg, Germany, in compliance with all relevant ethical regulations. Mice were housed under controlled conditions, namely 20–21 °C, 55–65% relative humidity, and 12:12 light–dark cycle. Food was available ad libitum for all animals. Animals were 6–8 weeks old when used. They were euthanized by carbon dioxide asphyxiation followed by cervical dislocation, and bone marrow, blood, and peritoneal lavage were harvested post mortem. For in vivo studies, mice were injected i.p. with 8 mg kg$^{-1}$ LPS (Sigma), and body temperature was monitored every 3 h for the duration of the experiment using an infrared thermometer (Bioseb). Mice were administered T863 (5 mg kg$^{-1}$, i.p.; SML0539-Sigma) or an equivalent volume of solvent (dimethylsulfoxide, control) 30 min prior to LPS injection. Serum and peritoneal lavage were collected at 2 h or 10 h post LPS injection.

**Primary cell cultures**. Bone marrow cells were grown in complete medium (RPMI-1640 medium containing 10 mM glucose, 2 mM ʟ-glutamine, 100 U ml$^{-1}$ penicillin–streptomycin and 10% FCS) with 20 ng ml$^{-1}$ murine macrophage colony- stimulating factor 1 (CSF-1; Peprotech) for 7 days, and supplemented with CSF-1 on days 3 and 5. On day 7, macrophages were harvested and then maintained in 20 ng ml$^{-1}$ CSF-1 for subsequent experiments in which they were either maintained in medium without any further additions (M0), or stimulated with 50 ng ml$^{-1}$ IFN-γ (R&D systems) and 20 ng ml$^{-1}$ LPS (M(γ + LPS)), or 20 ng ml$^{-1}$ IL-4 (Peprotech; MI(L-4)), for 18 h. In some experiments, cells were treated with 50 μM T863 (Sigma) and/or 10 μM PGE2 (Sigma) throughout the 18 h period.

**Lipidomics**. The protocol for lipid extraction was adapted from Matyash et al.[48]. Frozen cell pellets ($5 \times 10^5$ cells) were resuspended in ice cold PBS and transferred to glass tubes before the addition of methanol and methyl tert-butyl ether. The tubes were then shaken for 1 h at 4 °C. Water was added to separate the phases before centrifugation at $1000 \times g$ for 10 min. The upper organic phase was collected and dried in a Genevac EZ2 speed vac. Samples were resuspended in 2:1:1 isopropanol:acetonitrile:water prior to analysis. LC–MS was carried out using an Agilent Zorbax Eclipse Plus C18 column using an Agilent 1290 Infinity II UHPLC inline with an Agilent 6495 Triple Quad QQQ-MS. Lipids were identified by fragmentation and retention time, and comparison to standards, and were quantified using Agilent Mass Hunter software. Comparisons were made between relative amounts of lipid between conditions, extracted from equivalent cell numbers. Peak areas were quantile-normalized across the batch to generate the lipid intensities used for the plots and subsequent statistics shown in this manuscript.

**FA 20:4 analysis**. The extraction was done using the same method described on Lipidomics procedure, above. Data were acquired by LC–MS using an Agilent 1290 Infinity II UHPLC inline with a Bruker Impact II QTOF-MS operating in negative and positive ion mode (two injections per sample). Scan range was from 50 to 1600 Da. Mass calibration was performed at the beginning of each run. LC separation was on a Zorbax Eclipse plus C18 column ($100 \times 2$ mm, 1.8 μm particles) using a solvent gradient of 70% buffer A (10 mM ammonium formiate in 60:40 acetonitrile:water) to 97% buffer B (10 mM ammonium formiate in 90:10 2-propanol:acetonitrile). Flow rate was from 400 μL min$^{-1}$, autosampler temperature was 5 °C and injection volume was 2 μL. Data processing was carried out using Bruker MetaboScape 4 software. Fatty acid composition of TGs and Phospholipids was determined by matching measured spectra to reference spectra from LipidBlast. Peak height was used for relative quantification.

**Metabolite quantification**. Targeted metabolite quantification was carried out using the same LC–MS machine that was used for lipidomics. Samples were extracted in 50:30:20 methanol:acetonitrile:H$_2$O pre-cooled at −80 °C. Multiple reaction monitoring (MRM) settings were optimized for all metabolites using pure compounds. LC separation was on a Waters CSH-C18 column ($100 \times 2.1$ mm, 1.7 μm particles) using a binary solvent gradient of 100% buffer A (0.1% formic acid in water) to 97% buffer B (50:50 methanol:acetonitrile). Flow rate was 400 μL/min, autosampler temperature was 4 °C, and injection volume was 3 μL. Data processing was performed by an in-house R script. Peak areas were normalized to a fully $^{13}$C-labelled yeast extract (ISOtopic Solutions, Vienna).

**Palmitate tracing**. Cells ($3 \times 10^5$) were cultured in RPMI 1640, supplemented with 10 mM $^{13}$C-palmitate for 18 h, after which they were rinsed with cold 0.9% NaCl and extracted using 0.2 mL of 80% MeOH on dry ice, and dried using a SpeedVac. Label tracing was carried out using an Agilent 1290 Infinity II UHPLC inline with a Bruker impact II QTOF-MS operated in full scan (MS1) mode. LC parameters were identical to those used for targeted metabolite quantification. Data processing including correction for natural isotope abundance was performed by an in-house R script. Metabolite peaks were identified based on exact mass and matching of retention time to a pure standard.

**Glucose and glycerol tracing**. Cells ($3 \times 10^5$) were cultured in RPMI 1640, supplemented with 10 mM $^{13}$C-glucose for 18 h. Separately, media was supplemented with 10 mM $^{13}$C-glycerol for 30 min prior to collection of samples at 18 h post stimulation. Cells were rinsed with cold 0.9% NaCl and extracted using 0.2 mL of 80% MeOH on dry ice. Ten nanomolar norvaline (internal standard) was added. Following mixing and centrifugation, the supernatant was collected and dried using a SpeedVac. Dried extracts were analyzed using gas chromatography–mass spectrometry (GC–MS) (Agilent 5977). Correction for natural isotope abundance and calculation of fractional contribution was performed as described elsewhere[49].

**Lentiviral and retroviral production and cell transduction**. HEK293T cells were transfected using Lipofectamine 3000 (Thermo Fisher Scientific) with lentiviral packaging vectors pCAG-eco and psPAX.2 plus empty pLKO.1 control (EV) with a puromycin selection cassette (all obtained from Addgene) or a shRNA containing pLKO.1 targeting DGAT1 (GE Dharmacon CAT# RMM4534-EG13350-TRCN0000124791). Virus was collected from the supernatant of the cells. Bone marrow cultures were transduced in the presence of polybrene (8 mg ml$^{-1}$) on day 2 of culture. A 48 h, selection of transduced cells was performed with 6 μg ml$^{-1}$ puromycin (Sigma).

**RNA sequencing**. Total RNA was extracted with the RNAqueous-Micro Total RNA Isolation kit (Thermo Fisher Scientific) and quantified using Qubit 2.0 (Thermo Fisher Scientific), according to the manufacturer's instructions. Libraries were prepared using the TruSeq stranded mRNA kit (Illumina) and sequenced in a HISeq 3000 (Illumina) by the Deep Sequencing Facility at the Max Planck Institute for Immunobiology and Epigenetics. Sequenced libraries were processed with a pipeline optimized by the Bioinformatics core at the Max Planck Institute for Immunobiology and Epigenetics[50]. Raw mapped reads were processed in R (Lucent Technologies) with DESeq2[51] to determine differentially expressed genes and generate normalized read counts to visualize as heatmaps using Morpheus (Broad Institute).

**Flow cytometry**. In vitro stimulated macrophages were incubated in 5 μg ml$^{-1}$ anti-CD16/32 (#101302, Biolegend, 1:200), stained with Live Dead Fixable Blue (L23105,ThermoFisher, 1:500), and then surface stained with a fluorochrome-conjugated monoclonal antibody to F4/80 (BV421 #123131-Biolegend, clone BM8, 1:300), prior to fixation and permeabilization using BD Cytofix/Cytoperm kit (BD Biosciences) and staining with monoclonal antibodies to IL-6 (PE, #504503, Biolegend,clone MP5-20F3, 1:200) and pro IL-1β (FITC, #11-7114-82,eBioscience, clone NJTEN3, 1:200). Resting (M0) macrophages and macrophages stimulated with IFNγ + LPS, with or without DGAT1i 1 h priori to the addition of Brefeldin A (1:1000) were collected 5 h later. For ICS in cells treated with PGE2: cells were stimulated with IFNγ + LPS 1 h prior to tha addition of PGE2 (10 μM) and 1 h later Brefeldin A was added to the culture and ICS was performed after 5 h of subsequent culture. Peritoneal macrophages were incubated in 5 μg ml$^{-1}$ anti-CD16/32 (#101302, Biolegend, 1:200), stained with Live Dead Fixable Blue (L23105,ThermoFisher, 1:500), and then surface stained with a fluorochrome-conjugated monoclonal antibody F4/80 (APCcy7 #123117-Biolegend, clone BM8, 1:300), CD11b (PEcy7, #101215-Biolegend, clone: M1/70, 1:300) and Tim4 (APC, #129907-BioLegend, clone: F31-5G3, 1:200). For mitochondrial superoxide staining, ER staining, Mitotracker and membrane potential macrophages were incubated with 5 μM MitoSOX (#M36008, ThermoFisher), 1 μM ER tracker green or red (#E34251,#34250, ThermoFisher), 50 nM Mitotracker Deep Red (#M22426, ThermoFisher), 50 nM TMRM (#T668, ThermoFisher) in complete media without FCS for 10 min for MitoSOX staining and in complete media with FCS for 30 min for ER staining and Mitotracker. TMRM was incubated for 5 min. pHrodo™ Red *Staphylococcus aureus* Bioparticles™ (#A10010, ThermoFisher, 100μg ml$^{-1}$) was used for phagocytosis assays according to the manufacturer's instructions. BODIPY™ 493/503 (#D3922, ThermoFisher, 2 μM) and BODIPY® FL C16 (#D3821, ThermoFisher, 2μM) were used for neutral lipid staining and fatty acid uptake assays according to the manufacturer's instructions. Data were acquired by flow cytometry on an LSRII or LSR Fortessa (BD Biosciences) and analyzed with FlowJo v.10.1 (Tree Star).

**Confocal microscopy**. A total of $2 \times 10^5$ macrophages were plated on a 10 mm cover slip in a 24-well plate. Cells were subsequently cultured with or without γ + LPS for 18 h, followed by staining with Bodipy 493/503 (#D3922, ThermoFisher, 2 μM) for 1 h, fixation in 2% PFA for 20 min and washing with PBS. Slips were mounted with Vectashield with DAPI (#H1500-10, Vector Laboratories) and sealed

with nail polish before image acquisition using a Zeiss LSM 880 with Aryscan equipped with a 63× objective. Confocal images were analyzed and merged using ImageJ software.

**Electron microscopy**. A total of $5 \times 10^5$ macrophages were fixed in 2.5% glutaraldehyde (Sigma) in 100 mM sodium cocodylate (Sigma) and washed in cocodylate buffer. Following this step, samples were processed and imaged at the Electron Microscopy Laboratory at the University of Padova.

**ELISA**. Concentrations of IL-1β and IL-6 in cell culture supernatants or blood serum were measured using cytokine-specific ELISAs (BioLegend ELISA Max kits, #431301-IL-6 and #432603- IL-1β), according to the manufacturer's instructions. Absorbance was measured using a TriStar plate reader (Berthold Technologies). Standard curves were analyzed in Prism using second-order polynomial interpolation.

**ECAR measurements**. ECAR measurements were made with an XF-96 Extracellular Flux Analyzer (Seahorse Bioscience). A total of $1 \times 10^5$ BMDMs, were plated into each well of Seahorse X96 cell culture microplate and preincubated at 37 °C for a minimum of 45 min in the absence of $CO_2$ in unbuffered RPMI with 1 mM pyruvate, 2 mM L-glutamine and 25 mM glucose, with pH adjusted to 7.4. ECAR was measured under basal conditions. Results were collected with Wave software version 2.4 (Agilent).

**Quantitative PCR with reverse transcription**. RNA was isolated using the RNeasy kit (Qiagen) and single-strand cDNA was synthesized using the High Capacity cDNA Reverse Transcription Kit (Applied Biosystems). DGAT1 (Mm00515643_m1, Applied Biosystems) RT-PCR was performed with TaqMan primers using an Applied Biosystems 7000 sequence detection system. The expression levels of mRNA were normalized to the expression of β-actin (Mm02619580_m1, Applied Biosystems).

**Statistical analysis**. Statistical analysis was performed using prism 7 software (Graph pad) and results are represented as mean ± s.e.m. Comparisons for two groups were calculated using unpaired two-tailed Student's $t$ tests, comparisons of more than two groups were calculated using one-way ANOVA with Bonferroni's multiple comparison tests. We observed normal distribution and no difference in variance between groups in individual comparisons. Statistical significance: *$p <$ 0.05; **$p < 0.01$; ***$p < 0.001$. Further details on statistical analysis are listed in the figure legends.

**Reporting summary**. Further information on research design is available in the Nature Research Reporting Summary linked to this article.

## Data availability
RNA-sequence data have been deposited in Gene Expression Omnibus under the primary accession code GSE145523. The gene expression data set used in Fig. 1a and Supplementary Fig. 1a, b is available in Gene Expression Omnibus under accession code GSE123596 [https://www.ncbi.nlm.nih.gov/geo/query/acc.cgi]. The code used for palmitate tracing data processing is available from the authors upon request. The source data underlying Figs. 1b–g, 2a, c–r, 3a–g, i, j, l, and 4c–h and Supplementary Figs. 1a, b, d, f–g, 2a, e–g, j, k, m, 3a, b, f, i, k, and 4d, g–j are provided as a Source Data file. Source data are provided with this paper.

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

## Acknowledgements

We thank the members of the Pearce laboratories, the Electron Microscopy Laboratory at the University of Padova, and the FACS, Imaging, and Deep Sequencing facilities at the Max Planck Institute of Immunobiology and Epigenetics for technical support. This work was funded by by NIH (AI110481 to E.J.P.) and the Max Planck Society, and supported by the German Research Foundation (DFG) under Germany's Excellence Strategy (CIBSS EXC-2189 Project ID 390939984). A.C. is supported by the CAPES/Alexander von Humboldt Fellowship Foundation (88881.136065/2017-01), MC is supported by the Alexander von Humboldt Fellowship Foundation, M.M. is supported by Japan Society for the Promotion of Science and L.B.M. is supported by the Sao Paulo research foundation-FAPESP (2016/23328-0). Open access funding provided by Projekt DEAL.

## Author contributions

A.C. and E.J.P. designed research studies. A.C., L.B.M., N.V.T.B., M.C., A.M.C., F.H., M.M., G.C., R.I.K.G., and J.B. conducted the experiments. J.E.H. developed the lipid methodology. D.E.S. and N.R. performed bioinformatics analysis. E.L.P. and E.J.P. provided supervision and funding. A.C. and E.J.P. wrote the manuscript.

## Competing interests

E.J.P. and E.L.P. declare that they are Founders of Rheos Medicines. E.L.P. is a member of the Scientific Advisory Board of Immunomet Therapeutics. The other authors declare no competing interests.
