## [Peer Review File · Nature Communications]

Reviewers' comments:

Reviewer #1 (Remarks to the Author):

In this paper, Castoldi et al find evidence for a role for triglycerides in inflammatory macrophage activation. Inhibiting triglyceride synthesis blocked the production of cytokines and PGE2 and phagocytic activity. PGE2 production with downstream consequences such as IL1beta production seems especially important. These findings are interesting but the following issues need to be addressed.

1. What is the precise link between TGs and PGE2? This is missing here and is an important consideration. Do TGs lead to phospholipid synthesis with arachidonic then being released (which is likely) or is the arachidonic acid coming directly from the TGs? Without these kinds of data the link to PGE2 is somewhat indirect.
2. Is COX2 also being regulated? This might also contribute to the production of PGE2.
3. The link to IL1beta is clear here but why is IL6 also being regulated? Is that downstream of IL1?

Reviewer #2 (Remarks to the Author):

This manuscript by Castoldi and colleagues investigates the function of DGAT1-mediated triglyceride synthesis in macrophages during inflammation. The authors find that a proinflammatory stimulus upregulates lipid synthesis pathway genes, including DGAT1. Most intriguingly, they report that systemic inhibition of DGAT1 lowers inflammation, for instance measured as hypothermic response to LPS injection. They also find that the DGAT1 effect can be rescued in some assays by supplementation with prostaglandins. Overall, this area of research is interesting and some of the data are quite intriguing. However, there are at least two major problems that make it difficult to support publication of this study. First, throughout the manuscript, the treatment of cells or mice with a DGAT1 inhibitor is equated with lowering triglyceride synthesis and second there is no mechanistic understanding of the effects that are reported. Below these, and a number of less dominant, yet important points are explained in more detail:

- 1) In this paper, T863 treatment is equated with inhibiting TG synthesis. There are at least three problems with this assumption:
 - While it is reasonable to assume that inhibiting DGAT1 with this compound lowers TG synthesis, there are other effects and TG synthesis in many systems can be redundant (catalyzed by DGAT2). For instance, it is possible that specific lipid synthesis intermediates accumulate, triggering ER stress or mitochondrial dysfunction (independent of triglyceride synthesis per se). AS for the redundancy, it is puzzling why the authors have not included DGAT2 and DGAT1 /DGAT2 double inhibition in their studies (as this gene appears also upregulated).
 - Moreover, although T863 is well characterized, it is unclear whether it might have "off-target" effects. There are well characterized genetic models for DGAT1 deficiency (in cells, and animals) that could have been used to complement these studies.
 - It is hard to see how systemic administration of a drug can be linked to macrophages specifically. So, while I find the experiments presented e.g. in Fig. 2n very interesting, without a cell line-specific inactivation of DGAT1, they are next to impossible to interpret.
- 2) The mechanism of action for DGAT1 inhibition remains unclear. Is it through changes in gene expression, alterations of LD-localized biochemical reactions, or some other mechanism? The paper provides some correlations but does not really probe any mechanism. This would need to be clarified greatly in my opinion. In this regard, the experiments with PGE2 are interesting and Figure 2f is suggestive- but the link between DGAT1 inhibition and PGE2 synthesis remains unclear (and alternative explanations, not linked to TG synthesis but for instance ER or mito stress can be imagined).
- 3) there are some problems with the data presented: gene expression changes are not well statistically analyzed and the reported changes seem quite small. They also are somewhat

selective. For instance, the authors report all the GPAT genes, except GPAT1, which is thought to provide most substrate for triglyceride synthesis.

Similarly, the lipidomic and metabolomic data are somewhat weak: "peak area" is very arbitrary and hard to evaluate. Ideally, this is normalized to internal standards. For TG and some of the other lipids, other quantitative assays are easy to perform as well (in case they authors do not have access to lipidomics with standards).

For the point that IFN γ increases TG synthesis, ideally a pulse chase/experiment would be good to support the data in figure 1.

Moreover, cellular changes with LPS and DGAT1i are hard to evaluate (e.g. 3h), more data, and quantitative evaluation would need to be provided to make more clear insights about these phenotypes.

Point-by-point response to the reviewers' comments.

Revised sections are highlighted in yellow within the manuscript text file.

Reviewer #1:

In this paper, Castoldi et al find evidence for a role for triglycerides in inflammatory macrophage activation. Inhibiting triglyceride synthesis blocked the production of cytokines and PGE2 and phagocytic activity. PGE2 production with downstream consequences such as IL1 β production seems especially important. These findings are interesting but the following issues need to be addressed.

1. What is the precise link between TGs and PGE2? This is missing here and is an important consideration.

In early work, quantitative correlations between enhanced eicosanoid production and LD formation were established in different cell types, suggesting that LDs might function as specific sites involved in eicosanoid synthesis by cells engaged in inflammation¹⁻³. Later work then showed that PGE2 synthesis is localized to lipid droplets (LDs)⁴⁻⁶, and it was postulated that eicosanoid production is a major function of these organelles in immune cells⁷. Since LDs are stores of triacylglycerols (TGs), and inhibition of TG synthesis (using chemical inhibitors or genetic approaches to target DGAT1) resulted in loss of LDs and loss of PGE2 synthesis, we reasoned that the synthesis of TGs is necessary for PGE2 synthesis and proper macrophage inflammatory response.

To emphasize this in the revised paper, we have added the lines 59-61: *“Of relevance for this study, there is also a body of literature which demonstrates that LDs act as a platform for eicosanoid production from the lipid substrate, arachidonic acid^{7,8}.”*

In addition, we amended lines 245 to 247 of the manuscript to read: *“Given that LDs are both TG stores and sites for PGE2 synthesis, and loss of DGAT1 function resulted in loss of LDs and of IL-1 β production, we asked whether this was associated with reduced PGE2 levels.”*

2. Do TGs lead to phospholipid synthesis with arachidonic then being released (which is likely) or is the arachidonic acid coming directly from the TGs? Without these kinds of data the link to PGE2 is somewhat indirect.

We appreciate the Reviewer raising this point. We analyzed the contribution of fatty acid 20:4, arachidonic acid (FA 20:4 (AA)) to TGs and phospholipids by lipidomics in positive and negative ion mode by MS2 scan, and detected it in both. Of interest though, FA 20:4 (AA) increased in TG species in cells after activation with γ +LPS, and this increase was prevented by inhibition of DGAT1 inhibition (new Fig. 4d). In contrast, while we did measure FA 20:4 (AA) in phospholipids, the only significant change that we measured in this compartment was in phosphatidylcholine, as a decrease in activated vs. resting macrophages (new Fig. S4e). We interpret these findings as showing that inflammatory activation leads to an

increase in TGs containing AA, and that these comprise the substrate, located within LDs, for PGE2 synthesis.

We have included the following text in the revised paper to describe and discuss these findings (lines 249-255): “Moreover, we found that inflammatory macrophages build FA 20:4 (arachidonic acid, AA)-containing TG stores, which serve to focus substrate for PGE2 synthesis within these organelles. The inhibition of TG synthesis prevented the accumulation of this substrate store (Fig. 4d and Supplementary Fig. 4c). The cells also contained FA 20:4 (AA) in PC, PE, PG and PS (Supplementary Fig. 4d), but for the most part these were not dynamically regulated by activation (Supplementary Fig. 4d)” and (lines 313-317) “Our findings indicate that TG synthesis is necessary for the accumulation of AA-containing TGs to act as a substrate reservoir for PGE2 synthesis. In this scenario, coordinated increases in LD development and expression of genes related to positive regulation of prostaglandin-E synthase activity enable the framework for enhanced PGE2 synthesis,…”.

3. Is COX2 also being regulated? This might also contribute to the production of PGE2.

We found that the gene encoding COX2 (*Ptgs2*) was expressed more strongly in inflammatory vs. resting macrophages, as were other genes that regulate PGE2 synthesis, such as *Mgst1*, *Ptges* and *Nod2*. We have included this and additional relevant information related in the new Supplementary Figure 4b.

In addition, in lines 236-243, we now state: “PGE2 is produced by inflammatory macrophages, in which expression of prostaglandin synthase *Ptgs2* was elevated compared to in resting macrophages (Supplementary Fig. 4b). Furthermore, COX2 (encoded by *Ptgs2*) is upregulated in inflammatory macrophages compared to unstimulated macrophages (Supplementary Fig. 4b). In addition, expression of additional genes that regulate PGE2 synthesis, such as *Il1b*, *Ptges*, *Mgst1*, and *Nod2* were significantly upregulated, whereas *Ptgs1*, together with *Itgam* and *Cbr1* were downregulated (Supplementary Fig. 4b)”.

3. The link to IL1beta is clear here but why is IL6 also being regulated? Is that downstream of IL1?

IL-1 β is one of the most potent activators of IL-6 production⁹. Cellular responses to IL-1 β are mediated by cascades of intracellular events including activation of mitogen-activated protein kinases (MAPKs) involved in the activation of AP-1 and I κ B kinases (IKKs) involved in the activation of NF- κ B as well as Pi3kinase/AKT dependent pathways that will result in IL-6 transcription¹⁰. It is possible that decreased IL-6 expression and secretion observed in our model when DGAT1 function is lost is due to a decrease in IL-1 β production.

We have revised the paper on lines 154-156 to acknowledge this possibility: “IL-6 is known to be regulated by IL-1 β ¹¹ and thus reduced levels of IL-6 in these assays may be a downstream effect of inhibition of IL-1 β production”.

Reviewer #2

This manuscript by Castoldi and colleagues investigates the function of DGAT1-mediated triglyceride synthesis in macrophages during inflammation. The authors find that a proinflammatory stimulus upregulates lipid synthesis pathway genes, including DGAT1. Most intriguingly, they report that systemic inhibition of DGAT1 lowers inflammation, for instance measured as hypothermic response to LPS injection. They also find that the DGAT1 effect can be rescued in some assays by supplementation with prostaglandins.

Overall, this area of research is interesting and some of the data are quite intriguing. However, there are at least two major problems that make it difficult to support publication of this study.

First, throughout the manuscript, the treatment of cells or mice with a DGAT1 inhibitor is equated with lowering triglyceride synthesis and second there is no mechanistic understanding of the effects that are reported. Below these, and a number of less dominant, yet important points are explained in more detail:.

1- In this paper, T863 treatment is equated with inhibiting TG synthesis. There are at least three problems with this assumption:

a- While it is reasonable to assume that inhibiting DGAT1 with this compound lowers TG synthesis, there are other effects and TG synthesis in many systems can be redundant (catalyzed by DGAT2). For instance, it is possible that specific lipid synthesis intermediates accumulate, triggering ER stress or mitochondrial dysfunction (independent of triglyceride synthesis per se). As for the redundancy, it is puzzling why the authors have not included DGAT2 and DGAT1 /DGAT2 double inhibition in their studies (as this gene appears also upregulated).

The Reviewer's point is valid since, like DGAT1, DGAT2 can mediate the last step in TG synthesis¹²⁻¹⁴. We did find that both DGAT1 and 2 are expressed in inflammatory macrophages, but focused on the role of DGAT1 in TG synthesis because its expression was most strongly affected by activation with γ +LPS (New Fig.1a and Supplementary Fig. 1a). Our experiments show that loss of DGAT1 function has marked effects on LD and TG accumulation, and inflammatory macrophage activation. Thus, our data do not indicate a dominant role for DGAT2 in inflammatory activation, or suggest that it can substitute for DGAT1 function, at least when expressed at physiological levels.

The question of whether or not specific lipid intermediates accumulate and trigger ER stress or mitochondrial dysfunction is interesting. We did observe increased acylcarnitine accumulation as a result of inflammatory activation, and then additionally as a result of loss of DGAT1 function, and we speculate about what this indicates. However, we did not see direct evidence for increased ER stress or mitochondrial dysfunction (new Figs. 3m,k, and Supplementary Fig. 3d,h). In the figures, we include analysis of the genes related to positive regulation of response to endoplasmic reticulum stress such as *Xbp1*, *Ern1*, *Ddit3*, *Atf6*, *Hspa5*, *Eif2ak3*, *Atf4*, *Cebpb* and mitochondrial-unfolded-protein response genes *Dnaja3*, *Hspa9* and *Hspd1*, (Fig. 3m and S3h and Fig. 3k and S3d respectively). Expression of all of these genes was either unaltered or diminished as a result of loss of DGAT1 function.

Relevant to this, in the revised paper, we have added the following: lines 213-216:

“However, these changes were not indicative of increased mitochondrial stress, as

expression of mitochondrial-unfolded-protein response genes Dnaja3, Hspa9 and Hspd1 was not increased upon DGAT1 inhibition (Figs. 3k, Supplementary Fig. 3d) in LPS plus IFN γ -stimulated macrophages". And lines 224-228: "However, a lack of transcriptional response within the unfolded-protein response genes Xbp1, Em1, Ddit3, Atf6, Hspa5, Eif2ak3, Atf4, and Cebpb indicated that DGAT1 loss of function did not cause ER stress in inflammatory macrophages (Figs. 3m, Supplementary Fig. 3h)".

b- Moreover, although T863 is well characterized, it is unclear whether it might have "off-target" effects. There are well characterized genetic models for DGAT1 deficiency (in cells, and animals) that could have been used to complement these studies.

In addition to using the DGAT1 inhibitor T863, we performed genetic experiments using DGAT1 shRNA, and observed that suppressing DGAT1 expression using RNAi (Supplementary Fig. 2e) resulted in decreased LD and TG accumulation (Fig. 2g-h) and cytokine production (Figs 2j,l). We therefore feel reasonably confident that the observed effects are due to loss of DGAT1 function.

We address this in the paper on lines 283-292: *"Our data indicate that an activation induced increase in DGAT1 expression plays a dominant role in increased TG synthesis, while DGAT2 plays only a supportive role. Although we cannot exclude the possibility that the DGAT1 inhibitor T863 has off-target effects that impact inflammatory macrophage biology, data generated from experiments in which DGAT1 expression was suppressed by RNAi support the conclusion that DGAT1 is playing a critical role in the production of TGs and LDs in inflammatory macrophages. In both loss of function models, decreased TG synthesis and LD formation was associated with impaired inflammatory capacity. Our data are compatible with previous reports that DGAT1 inhibition is sufficient to reduce TG levels in macrophages and other cell types ^{15,16}.*

c- It is hard to see how systemic administration of a drug can be linked to macrophages specifically. So, while I find the experiments presented e.g. in Fig. 2n very interesting, without a cell line-specific inactivation of DGAT1, they are next to impossible to interpret.

The reviewer's point is valid. However, we took advantage of the in vivo model of LPS to induced sepsis to reproduce the in vitro data. There is a history in immunology of using the intraperitoneal LPS injection model of septic shock to address whether a given cell type or pathway is involved in regulating inflammation associated with sepsis. In this system there is a major peritoneal infiltration of macrophages and neutrophils ¹⁷, and these are believed to initiate and inflammatory response that leads to systemic disease which begin the inflammatory response. The value of this experiment is that the systemic administration of the drug had the expected effects on macrophages at the site of LPS injection – namely they had reduced LDs compared to cells from control mice injected with LPS alone, and this was associated with reduced systemic levels of inflammatory markers (cytokines such as IL-1 β and IL-6) that the rest of our data show are sensitive to DGAT1 inhibition in in vitro cultures of purified macrophages. Thus, we believe that the data from this in vivo model are valuable because they support the thrust of the findings from all of the other experiments which were performed on purified macrophages.

In attempt to address the reviewer's concern while keeping the data in the paper, we have reworked the paragraph lines 170 – 185 as follows (new text in blue italics). "We found that

inhibiting DGAT1 also had significant effects on inflammation in vivo. T863- treatment of mice with LPS-induced systemic inflammation significantly diminished disease severity, which was measured as the hypothermic response to i.p. LPS injection (Fig. 2n). Mice that had received the drug were active and appeared unaffected by LPS injection at the time of sacrifice (10 h post injection), whereas those that had received carrier alone exhibited sickness signs such as hunching, immobility and piloerection. *While in these experiments we cannot rule out systemic effects of T863 injection on cells other than macrophages*, post mortem analysis of peritoneal macrophages revealed lower bodipy 493/503 staining in resident (F4/80⁺ TIM4⁺) and recruited (F4/80⁺ TIM4^{neg}) macrophages (Figs. 2o, p), indicative of lower LD formation. *Thus i.p. T863 does impact the local macrophage response to LPS. Moreover, T863-treatment resulted in significantly decreased serum levels of the typical macrophage inflammatory products IL-1 β (Fig. 2q) and IL-6 (Fig. 2r). These results are supportive of our in vitro findings and taken together the data indicate that the synthesis of TGs and their storage in LDs supports inflammatory macrophage activation*".

2- The mechanism of action for DGAT1 inhibition remains unclear. Is it through changes in gene expression, alterations of LD-localized biochemical reactions, or some other mechanism? The paper provides some correlations but does not really probe any mechanism. This would need to be clarified greatly in my opinion. In this regard, the experiments with PGE2 are interesting and Figure 2f is suggestive- but the link between DGAT1 inhibition and PGE2 synthesis remains unclear (and alternative explanations, not linked to TG synthesis but for instance ER or mito stress can be imagined).

As discussed in detail above, we did not find convincing conventional evidence for either mitochondrial or ER stress associated with loss of DGAT1 function in inflammatory macrophages. Mechanistically, we think our data can be best explained by DGAT1 being important for TG synthesis, TG accumulation being central to LD development, and LDs being platforms for PGE2 production, which is itself important for IL-1 β production. Our new data showing that accumulated TGs in inflammatory macrophages contain arachidonic acid (Fig. 4d), the substrate for PGE2 synthesis, add weight to this argument.

3- There are some problems with the data presented: gene expression changes are not well statistically analyzed and the reported changes seem quite small. They also are somewhat selective. For instance, the authors report all the GPAT genes, except GPAT1, which is thought to provide most substrate for triglyceride synthesis.

In response to this concern, we reanalyzed the data to include all genes related to triglyceride biosynthetic process (from MGI). The revised gene set is shown in new Fig. 1a. Expression of *Gpat1* was not changed significantly as a result of stimulating macrophages with γ +LPS. However, with this re-analysis, we found that *Gpat3* expression was strongly increased in inflammatory macrophages (new Fig. 1a, new Fig S1b). We discuss these finding in the revised paper on Lines 98-101: "*Gpat3 which encodes the enzyme which catalyzes the conversion of glycerol-3-phosphate to lysophosphatidic acid in the synthesis of TG, was the TG synthesis pathway gene most upregulated as a result of inflammatory activation (Fig. 1a, Supplementary Fig. 1b)*".

4-Similarly, the lipidomic and metabolomic data are somewhat weak: "peak area" is very arbitrary and hard to evaluate. Ideally, this is normalized to internal standards. For TG and some of the other lipids, other quantitative assays are easy to perform as well (in case they authors do not have access to lipidomics with standards).

We thank the reviewer for making this point. In the revised paper we have included the mass spectrum for TGs (Supplementary Fig. 2b and Source Data file) and added the following statement (Lines 360-365) regarding lipidomics methodology: "*Lipids were identified by fragmentation and retention time, and comparison to standards, and were quantified using Agilent Mass Hunter software. Standards were used to identify the lipids. Comparisons were made between relative amounts of lipid between conditions, extracted from equivalent cell numbers. Peak areas were quantile normalized across the batch to generate the lipid intensities used for the plots and subsequent statistics shown in this manuscript*".

Quantile normalization is a simple and effective method to reduce variation from LC-MS-based lipidomics data, revealing the biological variance.

Moreover, we have now included in the Source Data file the representative Mass spectra for the metabolites we show in the manuscript.

5-For the point that IFN γ increases TG synthesis, ideally a pulse chase/experiment would be good to support the data in figure 1.

Sorry, but it is not clear here what the reviewer means when referring to a pulse/chase experiment.

6- Moreover, cellular changes with LPS and DGAT1i are hard to evaluate (e.g. 3h), more data, and quantitative evaluation would need to be provided to make more clear insights about these phenotypes.

We agree that apart from reductions in LDs, additional changes in LPS-activated cells were difficult to discern from the EM that we performed, and we were careful to not overinterpret the data that we did present. In reality we expected that we might see more significant changes in mitochondria due to acylcarnitine accumulation, but this was not the case. We hope to explore this further in the future.

References.

1. Bozza PT, Yu W, Penrose JF, Morgan ES, Dvorak AM, Weller PF. Eosinophil lipid bodies: specific, inducible intracellular sites for enhanced eicosanoid formation. *J Exp Med.* 1997;186(6):909-920.
2. Bozza PT, Payne JL, Morham SG, Langenbach R, Smithies O, Weller PF. Leukocyte lipid body formation and eicosanoid generation: cyclooxygenase-independent inhibition by aspirin. *Proc Natl Acad Sci U S A.* 1996;93(20):11091-11096.
3. Pacheco P, Bozza FA, Gomes RN, et al. Lipopolysaccharide-induced leukocyte lipid body formation in vivo: innate immunity elicited intracellular Loci involved in eicosanoid metabolism. *J Immunol.* 2002;169(11):6498-6506.
4. Accioly MT, Pacheco P, Maya-Monteiro CM, et al. Lipid bodies are reservoirs of cyclooxygenase-2 and sites of prostaglandin-E2 synthesis in colon cancer cells. *Cancer Res.* 2008;68(6):1732-1740.
5. D'Avila H, Melo RC, Parreira GG, Werneck-Barroso E, Castro-Faria-Neto HC, Bozza PT. Mycobacterium bovis bacillus Calmette-Guérin induces TLR2-mediated formation of lipid bodies: intracellular domains for eicosanoid synthesis in vivo. *J Immunol.* 2006;176(5):3087-3097.

6. Melo RC, D'Avila H, Wan HC, Bozza PT, Dvorak AM, Weller PF. Lipid bodies in inflammatory cells: structure, function, and current imaging techniques. *J Histochem Cytochem.* 2011;59(5):540-556.
7. Melo RC, Weller PF. Lipid droplets in leukocytes: Organelles linked to inflammatory responses. *Exp Cell Res.* 2016;340(2):193-197.
8. Bandeira-Melo C, Weller PF, Bozza PT. Identifying intracellular sites of eicosanoid lipid mediator synthesis with EicosaCell assays. *Methods Mol Biol.* 2011;717:277-289.
9. Tosato G, Jones KD. Interleukin-1 induces interleukin-6 production in peripheral blood monocytes. *Blood.* 1990;75(6):1305-1310.
10. Cahill CM, Rogers JT. Interleukin (IL) 1beta induction of IL-6 is mediated by a novel phosphatidylinositol 3-kinase-dependent AKT/IkappaB kinase alpha pathway targeting activator protein-1. *J Biol Chem.* 2008;283(38):25900-25912.
11. Dinarello CA. Interleukin-1 in the pathogenesis and treatment of inflammatory diseases. *Blood.* 2011;117(14):3720-3732.
12. Wilfling F, Haas JT, Walther TC, Farese RV. Lipid droplet biogenesis. *Curr Opin Cell Biol.* 2014;29:39-45.
13. Walther TC, Chung J, Farese RV. Lipid Droplet Biogenesis. *Annu Rev Cell Dev Biol.* 2017;33:491-510.
14. Chitraju C, Walther TC, Farese RV. The triglyceride synthesis enzymes DGAT1 and DGAT2 have distinct and overlapping functions in adipocytes. *J Lipid Res.* 2019;60(6):1112-1120.
15. Nguyen TB, Louie SM, Daniele JR, et al. DGAT1-Dependent Lipid Droplet Biogenesis Protects Mitochondrial Function during Starvation-Induced Autophagy. *Dev Cell.* 2017;42(1):9-21.e25.
16. Knight M, Braverman J, Asfaha K, Gronert K, Stanley S. Lipid droplet formation in Mycobacterium tuberculosis infected macrophages requires IFN- γ /HIF-1 α signaling and supports host defense. *PLoS Pathog.* 2018;14(1):e1006874.
17. Hotchkiss RS, Karl IE. The pathophysiology and treatment of sepsis. *N Engl J Med.* 2003;348(2):138-150.

REVIEWERS' COMMENTS:

Reviewer #1 (Remarks to the Author):

The authors have adequately addressed my concerns.

Reviewer #2 (Remarks to the Author):

The authors have made a good effort in addressing my comments. While I believe that a number of the key issues are not fully settled. Specifically, I feel it remains well possible that DGAT1 inhibition has an effect independent of TG production. Time (and additional experiments) will tell.

Yet, overall this manuscript is interesting, improved and I now support its publication.